# The Molecular and Function Characterization of Porcine MID2

**DOI:** 10.3390/ani13182853

**Published:** 2023-09-08

**Authors:** Jing Chen, Likun Zhou, Zhuosong Yang, Shijie Zhao, Wen Li, Yina Zhang, Pingan Xia

**Affiliations:** 1College of Life Science, Henan Agricultural University, Zhengdong New District Longzi Lake 15#, Zhengzhou 450046, China; chenjing@henau.edu.cn (J.C.); zlk2436@163.com (L.Z.); yzs15937216927@gmail.com (Z.Y.); 2Ministry of Education Key Laboratory for Animal Pathogens and Biosafety, College of Veterinary Medicine, Henan Agricultural University, Zhengdong New District Longzi Lake 15#, Zhengzhou 450046, China; zsj1002935527@163.com (S.Z.); lw839466399@163.com (W.L.)

**Keywords:** MID2, *Sus scrofa*, molecular characterization, function, immune

## Abstract

**Simple Summary:**

*Midline2* (*MID2*) is widely involved in development, innate immunity, cancer, viral infections, and disease, but its role in pigs remains unexplored. In this study, we describe the structure and function of porcine MID2. Our results revealed that the gene of pMID2 is highly conserved with human and other vertebrate animals and its expression is started by NF-κB. pMID2 is distributed in cell membrane and cytoplasm and widely expressed in pig lung, spleen, and other organs. Moreover, pMID2 could enhance the activity of the ISRE promoter, and is mainly involved in the JAK-STAT signaling pathway. The above findings provide useful information for studying the function of pMID2 in pigs.

**Abstract:**

Midline2 (MID2/TRIM1) is a member of the tripartite motif-containing (TRIM) family, which is involved in a wide range of cellular processes. However, fundamental studies on porcine MID2 (pMID2) are still lacking. In this study, we identified and characterized the full length MID2 gene of pig (*Sus scrofa*). The sequence alignment analysis results showed that pMID2 had an N-terminal RING zinc-finger domain, BBC domain, and C-terminal COS box, FN3 motif, and PRY-SPRY domain that were conserved and similar to those of other vertebrates. Furthermore, pMID2 had the highest expression levels in porcine lung and spleen. Serial deletion and site-directed mutagenesis showed that the putative nuclear factor-κB (NF-κB) binding site may be an essential transcription factor for regulating the transcription expression of pMID2. Furthermore, the immunofluorescence assay indicated that pMID2 presented in the cell membrane and cytoplasm. To further study the functions of pMID2, we identified and determined its potential ability to perceive poly (I:C) and IFN-α stimulation. Stimulation experiments showed pMID2 enhanced poly (I:C)-/IFN-α-induced JAK-STAT signaling pathway, indicating that pMID2 might participate in the immune responses. In conclusion, we systematically and comprehensively analyzed the characterizations and functions of pMID2, which provide valuable information to explore the pMID2 functions in innate immunity. Our findings not only enrich the current knowledge of MID2 in IFN signaling regulation but also offer the basis for future research of pig MID2 gene.

## 1. Introduction

Tripartite-motif (TRIM) proteins represent one of the largest classes of E3 ubiquitin ligases that regulate a variety of biological processes including development, cell proliferation, differentiation, signal transduction, tumor suppression, disease pathology, and innate immunity [1,2,3]. The TRIM family is ubiquitously expressed. With a long evolutionary history, the tripartite structure is highly conserved among metazoans, and it varies depending on the type of TRIM proteins [4]. The TRIM proteins are characterized by a conserved N-terminal, really interesting new gene (RING) domain, B-box motifs, and the coiled-coil domain, and additionally harbor a highly variable carboxyl-terminal domain in the C-terminal. The TRIM family has been subdivided into 11 distinct subgroups (C-I to C-XI,) according to the highly variable carboxyl-terminal domain [3]. Midline2 (MID2), known as TRIM1, is a member of the TRIM family. MID2 has an RBCC (RING-finger, B-boxes, and coiled-coil) motif in the N-terminal region, a subgroup one signature (COS) box, a fibronectin type III (FN3) repeat, and a SPIa and the ryanodine receptor (PRY-SPRY) domain in the C-terminal region as all C-I subfamily TRIM members [5]. While the role of FN3 domain and PRY-SPRY domain in MIDs proteins is unclear, the COS domain was shown to mediate MID2′s association with the microtubular apparatus [6]. MID2 and its homolog MID1 (known as TRIM18), either as homo- or hetero-dimerization, have been shown to regulate intracellular signaling to microtubules for promoting cell division [7].

MID2 plays an important role in tumor development, interacting physically with the breast cancer 1 early-onset gene (BRCA1) [8]. It has shown that MID2 was overexpressed in breast cancer and might act as a biomarker for prognosis [9]. Knockdown of MID2 significantly reduced the proliferation of MCF-7 and MDA-MB-231 cells in vitro and in vivo [9]. A recent study has revealed MORC4 could induce the expression of MID2 by recruiting STAT3 to MID2’s promoter region, leading to enhancement of the chemo-resistance to breast cancer cells [10]. Recently, MID2 mutation has been shown to cause X-linked developmental disorders and minor facial changes, including short philtrum, large ears, and a squint [11]. Taken together, these data suggested that MID2 plays an important role in tumor development and X-linked developmental disorders. Human MID2 has been reported to induce nuclear factor-κB (NF-κB) and activator protein 1 (AP-1) signaling, thus mediating regulation of immunity [12]. *Rhodeus uyekii* TRIM1/MID2 (RuTRIM1/MID2) has been shown to negatively regulate interferon-γ/lipopolysaccharide-induced nuclear NF-κB signaling, and may play a role in the inflammatory response [13]. These findings imply MID2 might be involved in the regulation of innate immunity. MID2 has been known to be expressed during development in humans, mice, and chicks [14,15,16]. However, although physiological and developmental functions of MID2 have yielded some information, the molecular characterization and cellular function of porcine MID2 (pMID2) is still unclear.

In this study, the full-length sequence of pMID2 from pigs was cloned and identified for the first time, and the molecular characterization was analyzed by bioinformatics. The phylogenetic analysis revealed that pMID2 was highly conserved among vertebrates and closely clustered with humans. Furthermore, gene expression analysis in 22 tissues from pigs showed that pMID2 had the highest expression levels in the porcine lung and spleen. Functional promoter analysis revealed that the region from −343 to +18 of the pMID2 promoter contained several positive regulatory elements and identified that NF-κB may be an essential transcription factor for regulating the transcription expression of pMID2. Additionally, we analyzed the subcellular localization and found pMID2 formed aggregates in the cell membrane and cytoplasm. Dual-luciferase reporter assays indicated that pMID2 could enhance the poly (I:C)-/IFN-α-induced JAK-STAT signaling pathway and might act as a positive regulator in immune responses. These results provide valuable information to investigate the pMID2 functions in the innate immune system, and contribute to explore other potential biological functions for this protein.

## 2. Materials and Methods

### 2.1. Cells and Tissues

CRL-2843 cells were cultured in RPMI 1640 supplemented with 10% fetal bovine serum (FBS) (Gibco, Invitrogen, Carlsbad, CA, USA), as previously described [17]. HEK293T cells were cultured in Dulbecco’s modified Eagle’s medium (DMEM) containing 10% FBS. PAMs were collected and cultured as previously described [18]. All cells were maintained in a humidified incubator with 5% CO_2_ at 37 °C.

Three six-week-old, healthy, large white Dutch Landrace crossbred piglets from local farms were killed to obtain the experimental tissues, including the heart, liver, spleen, lung, kidney, larynx, stomach, pancreas, thyroid, testis, thymus, skin, eye, muscle, bladder, trachea, brain, duodenum, jejunum, ileum, inguinal lymphaden, and mesenteric lymphaden. All the tissues were immediately snap-frozen in liquid nitrogen and stored at −80 °C. All porcine tissue collection procedures were performed according to protocols approved by the Animal Care and Use Committee of Henan Agricultural University (HNND2020112613, approval date: 26 November 2020).

### 2.2. Sequence, Structure, and Phylogenetic Analysis

The multiple sequence alignments of pMID2 homologues with those from mammals were carried out using CLUSTAL W [19] and the percentage identity of diverse sequences based on *Sus scrofa* was computed using DNAMan. A phylogenetic tree of pMID2 was constructed using the neighbor-joining method by MEGA-X [20] with 1000 bootstrap replicates. A prediction of molecular weight and isoelectric point of pMID2 were obtained using Expasy [https://web.expasy.org/cgi-bin/compute_pi/pi_tool (accessed on 30 July 2023)]. Secondary structure prediction of pMID2 protein was analyzed by PSIPRED server. Three-dimensional structure of pMID2 was performed by Swiss-Model [SWISS-MODEL.expasy.org (accessed on 20 May 2023)].

### 2.3. RNA Isolation and RT-qPCRs Analysis

Total RNAs were extracted from porcine tissues by an RNA extraction kit and reverse transcribed with the PrimeScript^TM^ RT reagent Kit with gDNA Eraser (TaKaRa), according to the manufacturer’s instructions. According to the subsequent RT-qPCR assay protocol, 1 µg of RNA was reverse transcribed into 20 µL of cDNA in each reaction system. The RT-qPCR was performed on a CFX96^TM^ real-time system using a SYBR green PCR mix according to the following protocol: pre-incubation at 95 °C for 30 s, followed by 40 cycles of amplification at 95 °C for 5 s and 60 °C for 34 s. Porcine glyceraldehyde 3-phosphate dehydrogenase (p-GAPDH) was used as the internal control to normalize the gene expression levels. The relative expressions of pMID2 mRNA were calculated using the 2^−ΔΔCT^ method. Table 1 lists all of the primers for RT-qPCR.

### 2.4. Plasmids Construction

The full-length sequence of pMID2 was amplified by PCR using the cDNA of the porcine lung. The PCR procedure was as follows: pre-incubation at 95 °C for 3 min, followed by 35 cycles of amplification at 95 °C for 15 s and 60 °C for 15 s and 72 °C for 1 min. The reaction was finally stopped after a 5 min extension at 72 °C. The PCR product was cloned into the mammalian expression vectors pCMV-Flag-N to generate Flag-pMID2. Various truncated plasmids of pMID2 were amplified from corresponding wild-type Flag-pMID2 construct using specific primers (Table 1) to generate Flag-pMID2-∆RING, Flag-pMID2-∆BBC, Flag-pMID2-∆COS, Flag-pMID2-∆FN3, and Flag-pMID2-∆SPRY. The pMID2 promoter, amplified from genomic DNA of porcine alveolar macrophages (PAMs), was cloned into pGL3-Basic to generate wild-type pMID2-Luc. Various truncated plasmids of pMID2 promoter were generated from corresponding wild-type constructs. NF-κB site mutant vectors of pMID2 were made by deleting the predicated NF-κB binding sites using the site-directed mutagenesis kit. The specific primers (Table 1) used for plasmid construction were designed by Primer Premier 5, and all plasmids were confirmed using DNA sequencing.

### 2.5. Cell Transfection and RNA Interference

The cells were seeded and grown in 6-well or 24-well plates until 80% confluence. Transient transfection of plasmids into CRL-2843 and HEK293T cells was then performed using Lipofectamine 2000 (Invitrogen) following the manufacturer’s instructions. Finally, they were subjected to dual-luciferase reporter and confocal microscopy analysis to detect the promoter activity and subcellular localization of pMID2, respectively. Small interfering RNAs (siRNAs) that targeted MID2 were synthesized by Gene Pharma. CRL-2843 cells were seeded in 6-well plates and transfected with 50 pM si-MID2 or negative-control siRNA (NC) using Lipofectamine 2000 according to the manufacturer’s instructions. The sequences of the siRNAs are listed on Table 2.

### 2.6. Immunofluorescence Microscopy

To further investigate the biological roles of pMID2, we analyzed the subcellular localization of pMID2 by immunofluorescence microscopy. HEK293T cells were plated on Nunc glass-bottom dishes with a confluency of 80% and then transfected with the indicated plasmids. At 24 h, the cells were fixed with 4% paraformaldehyde for 15 min and then blocked in 5% bovine serum albumin (BSA) for 1 h. After washing the cells thrice with phosphate-buffered saline and Tween 20 (PBST), they were incubated with appropriate primary antibody and secondary antibody. Images were acquired with a laser-scanning confocal microscope (Zeiss, Oberkochen, Germany).

### 2.7. Dual-Luciferase Reporter Assay

HEK293T cells were co-transfected with wild-type or truncated pMID2 promoter report plasmids, Renilla luciferase reporter plasmid (pRL-TK, used as an internal control). Twenty-four hours later, cells were lysed with 1× passive lysis buffer for dual-luciferase reporter assay (Promega), and the luminescent signal was measured by a Fluoroskan Ascent^TM^ FL Microplate Fluorometer (Thermo Scientific, Waltham, MA, USA) according to the manufacturer’s instruction. The plasmids of ISRE-Luc, pRL-TK were co-transfected with Flag-MID2 or an empty vector into CRL-2843 cells for 24 h, and then incubated with poly (I:C) or porcine IFN-α for 6 h. The resulting cell samples were processed as described above. All of the experiments were independently conducted in triplicates.

### 2.8. Statistical Analysis

The results are presented statistically as means ± SD. Student’s *t* test was used to determine the statistical significance of differences between experimental groups. Statistical analyses were performed using the GraphPad Prism 8.3.0 software. The criteria for statistical significance and high significance were *p* < 0.05 and *p* < 0.01, respectively.

## 3. Results

### 3.1. Phylogenetic and Sequence Analysis of pMID2

To study the molecular evolution and analyze phylogenetic relationships of pMID2 to its homologous proteins in other species, we constructed a phylogenetic tree of MID2 based on amino acid sequences with MEGA-X software (10.1.8) (Figure 1). We analyzed 23 species, including *Danio rerio*, *Cricetulus griseus*, *Myotis brandtii*, *Myotis davidii*, *Xenopus tropicalis*, *Canis lupus familiaris*, *Rattus norvegicus*, *Mesocricetus auratus*, *Thunnus albacares*, *Mustela putorius furo*, *Heterocephalus glaber*, *Chelonia mydas*, *Nannospalax galili*, *Camelus ferus*, *Macaca mulatta*, *Larimichthys crocea*, *Gallus gallus*, *Felis catus*, *Bos taurus*, *Homo sapiens*, *Mus musculus*, *Anas platyrhynchos*, and *Sus scrofa*. The phylogenetic analysis revealed that pMID2 was closely clustered with homologues from other vertebrates, suggesting that pMID2 shares a common ancestor in 23 species.

The full length of pMID2 was cloned from the porcine lung by RT-PCR. The coding sequence (CDS) of pMID2 consists of 2148 bp, which was predicted to encode a 715 amino acid protein. The deduced pMID2 protein has a theoretical pI of 7.24 and was calculated to have a molecular mass of approximately 81.15 kDa. The NCBI Conserved Domain prediction results revealed that pMID2 has an N-terminus RING zinc finger domain (position 1 to 70), BBC (including B-box1, B-box2, and coiled-coil domain) (position 219–344) and C-terminus COS domain (position 323–374), and FN3 domain (position 381–511), PRY/SPRY domain (position 516–684) (Appendix A). Additionally, sequence analysis demonstrated that pMID2 (*Sus scrofa*) shared 99.16% sequence similarity with the MID2 of *Homo sapiens* (NP_001369680.1) and had 99.44% and 92.79% similarities with MID2 in *Macaca mulatta* (XP_014983453.1) and *Mus musculus* (NP_001345295.1), respectively (Appendix A). In conclusion, these results showed that MID2 has a high alignment in the four species and shares a common structural domain.

### 3.2. Structural Analysis of pMID2

Subsequently, we predicted the secondary structure of pMID2 using PSIPRED. As shown in Figure 2A, the predicted pMlD2 protein was composed of 27.8% α-helical structures and 20.45% extended strand. Moreover, to further investigate the structure of pMID2, domain and three-dimensional structure prediction was conducted using the NCBI Conserved Domain Database and the Swiss Model (Figure 2B).

### 3.3. Tissue Distribution and Expression Analysis of pMID2

To characterize the expression of pMID2 mRNA in different tissues, total RNAs were extracted from the porcine heart, liver, spleen, lung, kidney, larynx, stomach, pancreas, thyroid, testis, thymus, skin, eye, muscle, bladder, trachea, brain, duodenum, jejunum, ileum, inguinal lymphaden, and mesenteric lymphaden and subjected to RT-qPCR analysis. As shown in Figure 3, pMID2 was generally expressed in all the detected porcine tissues. The mRNAs of pMID2 in the lung, spleen, kidney, skin, duodenum, pancreas, heart, liver, stomach, jejunum, brain, eye, and larynx displayed a higher expression, while the mRNAs in the ileum, trachea, thymus, inguinal lymphaden, bladder, mesenteric lymphaden, testis, muscle, and thyroid showed the least amount of expression. These data indicated that pMID2 was ubiquitously expressed in all the tissues examined in pigs. The tissue distribution of pMID2 expression was abundant in lung and spleen, which are the two important immune-related tissues. The results imply that pMID2 might have a role in immune processes.

### 3.4. Functional Promoter Analysis of pMID2

In order to define the promoter region that is critical for the expression of the pMID2 gene, several promoter deletion fragments were generated and cloned into pGL3-basic vector: −1852/+18, −1366/+18, −890/+18, −444/+18 (Figure 4A). The result of dual-luciferase reporter assay showed deletion of the region from −1852 to −890 did not obviously influence the activation of pMID2 promoter, whereas the −444/+18 region resulted in an almost 60% decrease of the promoter activity, suggesting the −444/+18 regions contained several positive regulatory elements for the expression of pMID2. To find the essential *cis*-regulatory elements of the pMID2 promoter, the fragments from positions −343, −235, and −152 to +18 were constructed in the pGL3-basic vector, and the promoter’s activation was tested in HEK293T cells. The results showed that the fragments from position −343 to −235 were important for the activation of the pMID2 promoter (Figure 4B), which suggests that the region between −343 and −235 in the pMID2 promoter was an important *cis*-activating element for the expression of pMID2.

Next, to further explore which *trans*-factors played important roles in the transcription of pMID2, we analyzed the region of the pMID2 promoter from position −343 to −235 using the PROMO program [20], and found this region included one NF-κB binding site. HEK293T cells were treated for 1 h with the NF-κB inhibitor (BAY11-7082), and then were transfected with the reporter plasmid pMID2. The results of luciferase activity assays showed that the NF-κB inhibitor exerted an inhibitory effect on the activation of the pMID2 promoter (Figure 4C), suggesting that NF-κB was involved in the regulation of pMID2 expression. To further examine the roles of NF-κB in activation of pMID2 promoter, we subsequently generated a mutation of the NF-κB binding site and found that the NF-κB was important for transcriptional expression of the pMID2 (Figure 4D). Thus, the results described above suggest that NF-κB was a critical transcriptional factor for the expression of pMID2.

### 3.5. Cellular Localization of pMID2

To further visualize the subcellular distribution of pMID2, we transfected the Flag-MID2 plasmid into HEK293T cells and then analyzed them by confocal fluorescence microscopy. The results revealed that the red fluorescent signals of Flag-pMID2 were distributed in the cell membrane and cytoplasm (Figure 5B). To examine the effect of pMID2 structural domain on the intracellular localization of the protein, we designed and constructed the following truncated plasmids (Figure 5A): Flag-pMID2-∆RING, Flag-pMID2-∆BBC, Flag-pMID2-∆COS, Flag-pMID2-∆FN3, Flag-pMID2-∆SPRY, and then these constructs were transfected into HEK293T, respectively. Notably, deletion of N-terminal RING and BBC domains showed the pMID2 protein was also localized in the cell membrane and cytoplasm, while deletion of C-terminal COS, FN3, and SPRY domains indicated the pMID2 protein was primarily distributed in the cytoplasm. This finding suggested that deletion of N-terminal RBCC domains had no effect on the localization of pMID2 protein; however, deletion of the C-terminal each domain of pMID2 showed remarkable changes in subcellular localization.

### 3.6. Positive Regulatory of JAK-STAT Signaling Pathway by pMID2

To demonstrate the effect of pMID2 on interferon-stimulated response element (ISRE) promoter activity, CRL-2843 cells were co-transfected with an empty vector or pMID2 expression plasmid together with the porcine ISRE luciferase reporter plasmid and pRL-TK. After 24 h transfection, the cells were treated with poly (I:C) for 6 h (Figure 6A). Luciferase activity assay indicated that the pMID2 remarkably enhanced poly (I:C)-induced ISRE promoter activity. Furthermore, we found that pMID2 dramatically enhanced ISRE activation in a pMID2-dose-dependent manner (Figure 6C). To further investigate the function of pMID2 in ISRE promoter activity, synthesized siRNA targeting pMID2 was used to suppress endogenous MID2 expression in CRL-2843 cells. As shown in Figure 6B, the knockdown of pMID2 significantly reduced ISRE activation. Collectively, these results indicated that pMID2 could significantly increase poly (I:C)-induced ISRE production. Similarly, the above experiments were conducted by IFN-α treatment and the result showed that pMID2 could also promoted IFN-α-induced the ISRE activation (Figure 6D–F). Taken together, these findings indicate positive regulation of poly (I:C)/IFN-α-induced ISRE activation by pMID2, and the promotion effect of pMID2 was mainly involved in the JAK-STAT signaling pathway.

## 4. Discussion

To the best of our knowledge, our study represents the first to describe the molecular characterization and functionality of porcine MID2, primarily including phylogenetic study, sequences analysis, predicted structures, expression profiles, functional promoter analysis, subcellular localization, and potential biological function. The phylogenetic analysis revealed that the pMID2 was positioned on a branch of the mammalian subcluster and was most similar to human. Multiple sequence alignment demonstrated that the amino acid sequence of pMID2 (*Sus scrofa*) shared high identity with those of *Homo sapiens*, *Macaca mulatta* and *Mus musclus*, suggesting they might share conserved functions. Moreover, our alignment analysis revealed that the RING domain, BBC domain, COS box, FN3 motif, and PRY/SPRY domain were conserved among the MID2 sequences of all species examined. A long alpha helix was predicted in secondary and shown in tertiary structures, which were also seen in RING-type E3 ubiquitin ligase, as we know, demonstrating this helix may play a significant role in ubiquitination. These residues are important for the structure and function of TRIM proteins and have been reported as invariant among TRIM family members. Taken together, our findings demonstrated that pMID2 has conserved structural features and close phylogeny with MID2 orthologs.

In our study, the pMID2 was ubiquitously expressed in tissues examined and had the highest expression levels in porcine lung and spleen. MID2 in other species has been also reported to be widely expressed [14,21], whereas different species of MID2 display diverse expression patterns. For instance, *Homo sapiens* MID2 displays low embryonic expression in kidney and lung; however, in adult tissues, MID2 was predominantly expressed in prostate, followed by ovary and smooth muscle [14]. In contrast, in mice, MID2 was expressed early in the developing central nervous system (CNS), kidney and heart, while a significant level of expression was found in thymus, lung, and thyroid at adult stage [14]. By contrast, the expression of *Xenopus* MID2 was undetectable at neurula stages, but it had a weak expression in the pineal gland, otic vesicle, and heart tube at the tailbud stages [21]. *Rhodeus uyekii* (RuTRIM1/MID2) was abundant in hepatopancreas and spleen, closely related to immune system, suggesting Ru TRIM1/MID2 might have a role in fish immune system [13]. Possible explanations for the different expression patterns among species include intra-species variation and divergence during developmental stages. Taken together, the tissue expression profile showed that pMID2 had a broad ectopic expression in lung and spleen, and immune-related tissues/organs, which provided the tissue-level spatial fundamentals of the pMID2 gene as a pleiotropic immune regulator in porcine immune system. It is a limitation that the analysis of the expression profile of pMID2 was conducted with only three pigs; however, further research is ongoing and will confirm this result. Additionally, further elucidation of the expression profile of pMID2 in porcine tissues will contribute to uncover the function of pMID2 in the pig immune system.

The pMID2 promoter sequence from −343 to +18 was identified as an essential regulatory element. Further findings suggested that NF-κB may be a critical transcription factor for regulating MID2 expression. Our findings offer new information that point out a link between the NF-κB transcription factor and the pMID2 gene, which contribute to further investigation into the transcriptional regulation mechanism of pMID2. This study provides firstly evidence that the pMID2 promoter has an essential transcriptional regulation pattern characterized by multiple positive regulation elements. This transcriptional regulation pattern was consistent to its wide expression, suggesting that pMID2 may participate in different signal pathways, achieving multiple functions.

Although human MID2 was originally observed to be associated with the microtubule network and localize with microtubules [7,14], our study showed pMID2 was localized in the cell membrane and cytoplasm. The results were consistent with the RuTRIM1/MID2 that formed aggregates in cytoplasmic bodies [13]. These differences may be related to the differences in sequence variation among the species, cellular environments, or specific functions. In the TRIM family, several domains affect the subcellular localization of the proteins [22]. Our study revealed that the RING finger domain and BBC domain did not affect the localization of pMID2, whereas the deletion mutants of the COS domain, FN3 domain, and SPRY domain showed the localization of pMID2 changes from the cell membrane to the cytoplasm. Similar results were obtained from RuTRIM1/MID2 that deletion of the RING finger domain did not affect the localization of the RuTRIM1/MID2 proteins compared to the full-length RuTRIM1/MID2 [17]. Deletion of the C-terminal domain of *Homo sapiens* MID2 has been shown to disrupt the normal subcellular distribution of the MID2 protein, resulting in dissociation from the microtubules with the formation of cytoplasmic clumps [23]. These results indicate that each domain of the MID2 protein has its own role in subcellular localization.

The JAK/STAT pathway is known to be activated in interferon (IFN)-mediated antiviral activity. Reports have indicated that some TRIMs could regulate the JAK/STAT signaling pathway. For example, TRIM6 has been shown to modulate the IFNα/β-induced JAK/STAT signaling pathway by mediating STAT1 phosphorylation [24,25]. TRIM59 interacts with STAT1 by recruiting PIAS1 (protein inhibitor of activated STAT 1) to suppress the STAT1 activation, and it also suppresses IL-1β-induced activation of the JAK2/STAT3 pathway [26,27]. TRIM8 interacts with the negative regulatory factor, SOCS-1 (suppressor of cytokine signaling 1), PIAS3 (protein inhibitor of activated STAT 3), and Hsp90-β (heat shock protein 90-β) and mediates their degradation, allowing the activation of the JAK-STAT pathway [28,29,30]. It has been consistent in our findings that pMID2 significantly enhanced poly (I:C)-/IFN-α-induced activation of the JAK-STAT signaling pathway. Certainly, there is a need to further explore the detailed regulatory mechanism between pMID2 and the JAK-STAT signaling pathway.

Previous studies have reported some TRIM family members closely relate to specific tumorigenic pathways and cellular stress response regulation, indicating their diverse contributions to tumor development [31,32]. For example, TRIM14, TRIM27, TRIM29, and TRIM52 have been reported to mediate the overactivation of STAT3 and induce the expression of downstream target genes, such as MMP-2 (matrix metalloproteinases 2), MMP-9, and VEGF (vascular endothelial growth factor), thus promoting cancer cell migration and invasion [33,34,35,36]. Similarly, a high level of MID2 expression was significantly correlated with breast cancer progression [9]. A recent study revealed that MORC4 might confer increased chemoresistance to breast cancer cells via STAT3-mediated MID2 upregulation [10]. It is intriguing to investigate the effects of MID2 on breast cancer via the JAK-STAT signaling pathway. Malignant tumors are rare in pigs, but lymphosarcoma, nephroblastoma, and melanoma are the most common [37]. Since MID2 has been involved in human tumor development and acts as a prognostic biomarker, pMID2 might serves as a neoplastic biomarker in tumors of pigs. However, the mechanism by which pMID2 regulates tumors in pigs needs to be further explored. Other studies point to MID2 having broader biological roles; for example, MID2 has been found to exhibit antiretroviral activity against N-tropic murine leukemia virus (MLV) by inhibiting the late stages of virus replication as well as capsid-specific restriction activity [12]. Clinically, MID2 also was implicated in genetic developmental disorders [11,38]. Taken together, these results revealed that MID2 is a multifunctional protein, associated with development, innate immunity, cancer, viral infections, and disease. The future dissection of the biological function of MID2 will be necessary to form a comprehensive picture and for future clinical application.

## 5. Conclusions

In summary, we have identified, cloned, and characterized the pMID2 from pigs. The finding was that pMID2 is primarily expressed in the porcine lung and spleen, immune-related tissue/organ, and pMID2-enhanced poly (I:C) and IFN-α-induced JAK-STAT signaling pathway implicating pMID2, and might act as a positive regulator in immune responses. These results support the theory that pMID2 could be considered as a candidate for enhancing immunity and antiviral effects in pigs. Broad-spectrum antivirals can be developed with this knowledge, thereby contributing to the development of novel antiviral therapies and innovative strategies for combating viral infections in the pig industry.

## Figures and Tables

**Figure 1 animals-13-02853-f001:**
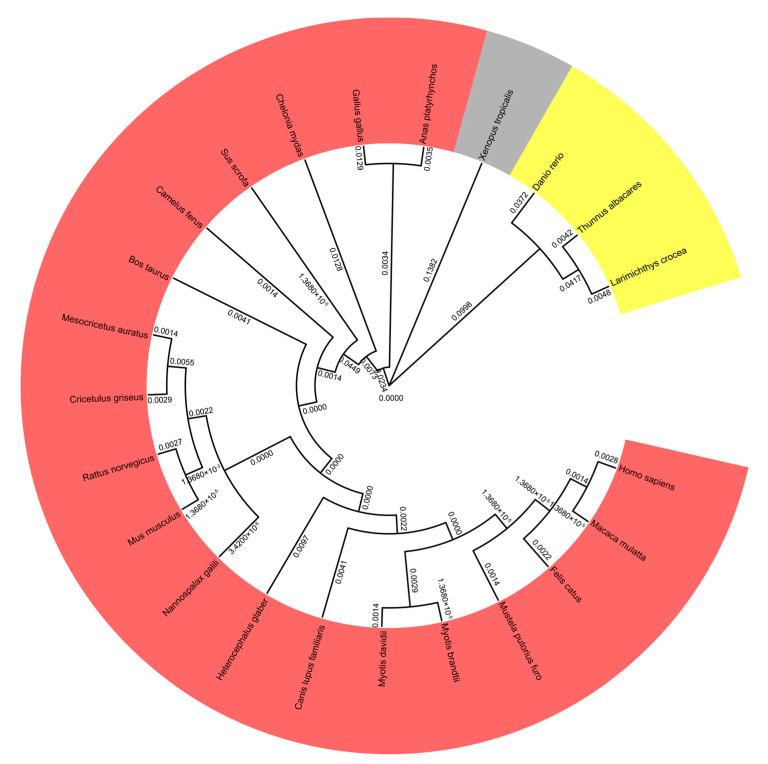
Phylogenetic analysis of pMID2. A neighbor-joining tree was constructed based on the amino acid sequences of pMID2 from different species using MEGA-X software. The numbers beside the branches represent genetic distance (GD). All of the GD values are less than 1, suggesting that MID2 has a high similarity in all these 23 species.

**Figure 2 animals-13-02853-f002:**
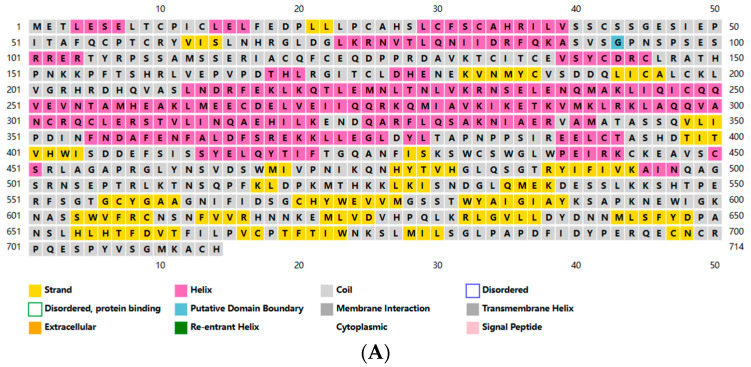
Structure analysis of pMID2. (**A**) Secondary structure prediction of pMID2 proteins using PSIPRED server. The α-helix region is marked in pink, β-strand in yellow, and extended coil region in gray. The predicted pMlD2 protein has a 27.8% α-helical structures and 20.45% extended strand according to GOR4 [https://npsa-prabi.ibcp.fr/cgi-bin/npsa_automat.pl?page=npsa_gor4.html (accessed on 30 July 2023)] prediction. (**B**) Three-dimensional structure of pMID2 was performed by Swiss Model [SWISS-MODEL.expasy.org (accessed on 20 May 2023)]. Three-dimensional structure is colored as yellow, magenta, blue, red, orange, wheat, which represent RING-HC_MID2, Bbox1_MID2_C-I, Bbox2_MID2_C-I, BBC, FN3, SPRY_PRY_MID2 domains in order. COS domain overlaps with BBC; COS domain is shown at right as brown. Homology models with best quality scores were selected and visualized with PyMOL program.

**Figure 3 animals-13-02853-f003:**
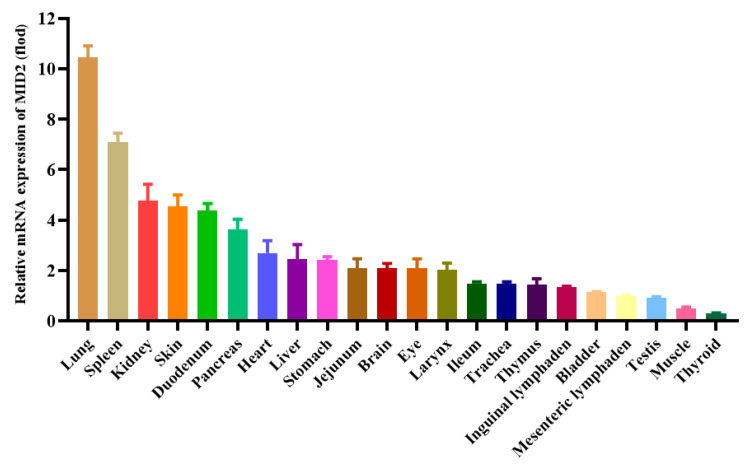
Relative pMID2 mRNA expression in different tissues. The relative mRNA expression levels of pMID2 were detected by RT-qPCR and were compared against gene expression level in thyroid. The tissues were ordered according to relative expression levels from highest to lowest. Error bars represent SD.

**Figure 4 animals-13-02853-f004:**
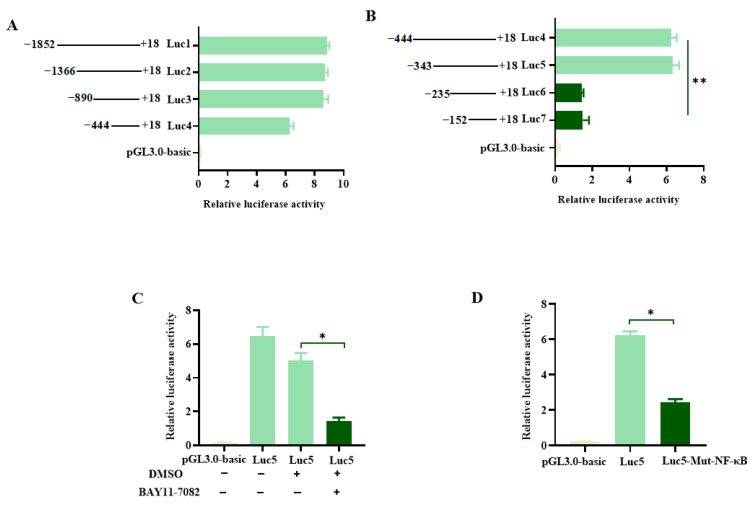
The 5′-deletion analysis of the promoter activity of pMID2 and site-directed mutation of an important cis-element. (**A**,**B**) HEK293T cells were co-transfected with different truncated pMID2 promoter report plasmids and pRL-TK, and 48 h later the cells were harvested for luciferase assays. (**C**) HEK293T cells were treated with DMSO or NF-kB inhibitor (BAY11-7082;) for 1 h, and followed by co-transfection of pMID2 promoter report plasmid (Luc5) or pGL3.0-basic with pRL-TK. Forty-eight hours later, cells were harvested for luciferase assay. (**D**) HEK293T cells were co-transfected with pMID2 promoter report plasmid (Luc5) or mutant vector (Luc5-Mut-NF-κB) along with pRL-TK or pGL3.0-basic for 48 h and then harvested for luciferase assay. We found 1870 bp of the 5′ promoter region of pMID2 (−1852 to +18 corresponds to the transcription initiation site) was linked to the luciferase gene, designated −1852/+18 (Luc1). Serial constructs in which the promoter region was deleted are depicted. The resultant plasmids were named −1366/+18 (Luc2), −890/+18 (Luc3), and −444/+18 (Luc4). Serial 5′-deletion constructs generated for −444/+18. The resultant plasmids were named −343/+18 (Luc5), −235/+18 (Luc6), and −152/+18 (Luc7). Mutation of a *cis*-element on −343/+18 (Luc5-Mut-NF-κB), which was predicted to be a NF-κB binding site. Error bars: means ± SDs of 3 independent tests. Student’s *t*-test: * *p* < 0.05; ** *p* < 0.01 compared to control.

**Figure 5 animals-13-02853-f005:**
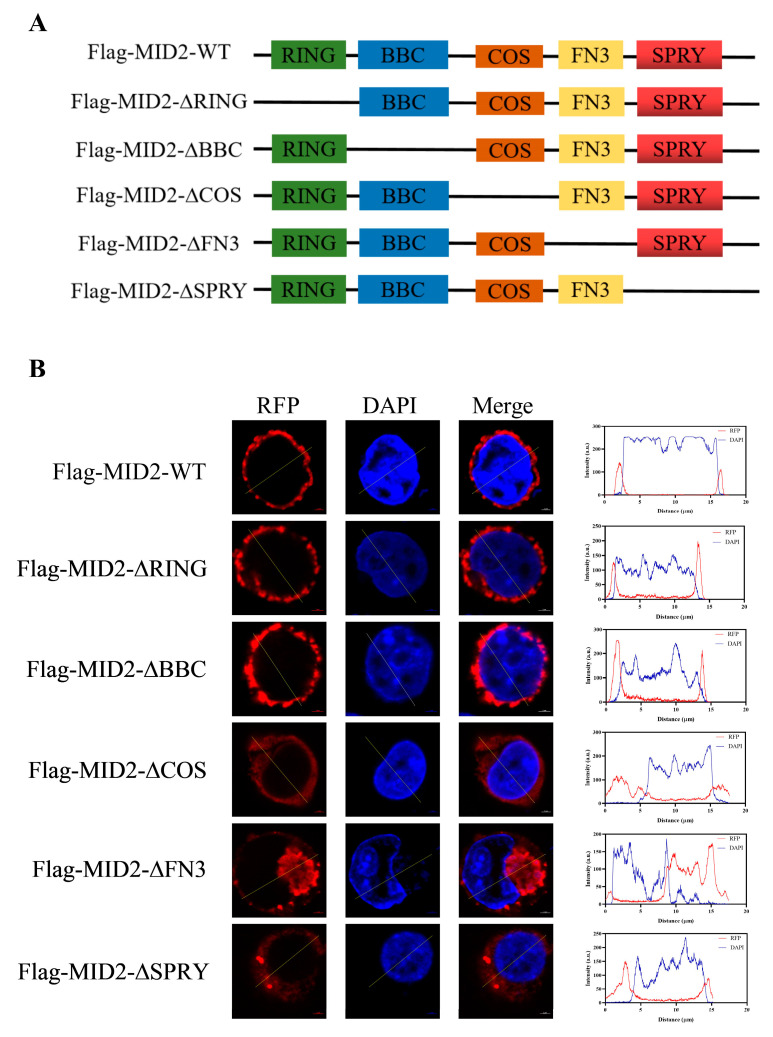
Subcellular localization analysis of pMID2. (**A**) Schematic representation of pMID2 and its mutants. RING: N-terminal really interesting new gene domain, BBC: N-terminal two B-box domain and one coiled-coil domain, COS: C-terminal subgroup one signature domain, FN3: C-terminus fibronectin type 3, SPRY: C-terminus SPIa and the ryanodine receptor domain. (**B**) HEK293T cells were transfected with Flag-pMID2-WT, Flag-pMID2-∆RING, Flag-pMID2-∆BBC, Flag-pMID2-∆COS, Flag-pMID2-∆FN3, Flag-pMID2-∆SPRY for 48 h, respectively. Cells were fixed and observed under a confocal microscopy; the cells were imaged for pMID2 (red). Their nuclei were stained with DAPI. Scale bar: 5 µm.

**Figure 6 animals-13-02853-f006:**
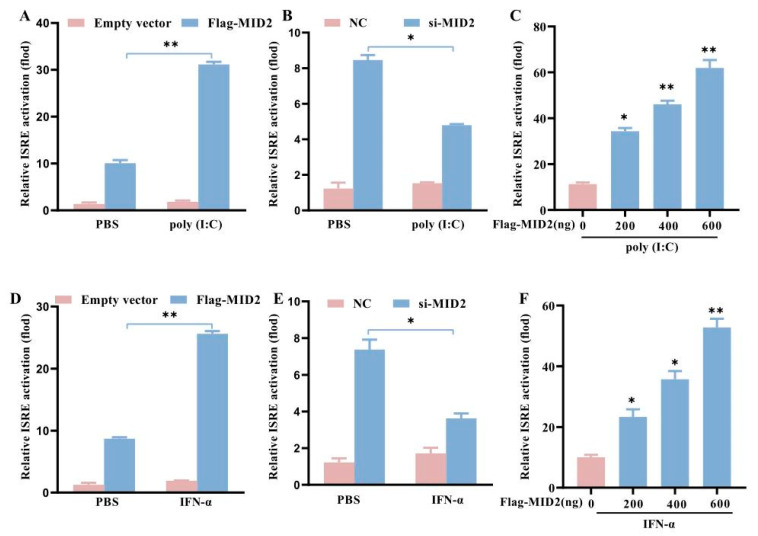
pMID2 enhances poly (I:C)/IFN-α-induced ISRE promoter activity. (**A**,**D**) ISRE promoter reporter vectors along with pRL-TK were co-transfected with Flag-MID2 or empty vector into CRL-2843 cells for 24 h, followed by incubation with poly (I:C) (10 μg/mL) (**A**) or porcine IFN-α (10 U/mL) (**D**) for 6 h. Then, cells were harvested for luciferase assay. (**B**,**E**) CRL-2843 cells were transfected with the NC, or si-MID2, and ISRE promoter reporter vectors along with pRL-TK for 24 h and then stimulated with poly (I:C) (10 μg/mL) (**B**) or porcine IFN-α (10 U/mL) (**E**). Cells were collected 6 h later for luciferase assay. (**C**,**F**) ISRE promoter reporter vectors along with pRL-TK were co-transfected with different concentrations of Flag-MID2 or empty vector into CRL-2843 cells for 24 h, followed by incubation with poly (I:C) (10 μg/mL) (**C**) or porcine IFN-α (10 U/mL) (**F**) for 6 h. Then cells were harvested for luciferase assay. Error bars: means ± SDs of 3 independent tests. Student’s *t*-test: * *p* < 0.05; ** *p* < 0.01 compared to control.

**Table 1 animals-13-02853-t001:** Primer sequences for gene cloning and RT-qPCR.

Primer Name	Sequence (5′-3′)	Purpose
Flag-pMID2-For	AAGCTTATGGGTGAAAGCCCAGCC	expression
Flag-pMID2-Rev	GTCGACCTAGAGAGCCGGTGAATTATATGCT	
Flag-pMID2-∆RING-For	AAGCTTAGGTATGTTATCTCGCTGAACCACC	
Flag-pMID2-∆RING-Rev	GTCGACCTAGAGAGCCGGTGAATTATATGCT	
Flag-pMID2-∆BBC-For	AAGCTTATGGGTGAAAGCCCAGCC	
Flag-pMID2-∆BBC-Rev	GTCGACCTAGAGAGCCGGTGAATTATATGCT	
Flag-pMID2-∆COS-For	AAGCTTATGGGTGAAAGCCCAGCC	
Flag-pMID2-∆COS-Rev	GTCGACCTAGAGAGCCGGTGAATTATATGCT	
Flag-pMID2-∆FN3-For	AAGCTTATGGGTGAAAGCCCAGCC	
Flag-pMID2-∆FN3-Rev	GTCGACCTAGAGAGCCGGTGAATTATATGCT	
Flag-pMID2-∆SPRY-For	AAGCTTATGGGTGAAAGCCCAGCC	
Flag-pMID2-∆SPRY-Rev	GTCGACTTAGTCAATGAATATATTTCCTGCTGC	
Luc1(−1852/+18)-For	GGTACCGTTATTGGATTGCAGTTCTTTACG	promoter
Luc1(−1852/+18)-Rev	AAGCTTTTTACCCGGAGCATTCCCG	
Luc2(−1366/+18)-For	GGTACCCCATGTTCTGAAAAGAATCTCCTCT	
Luc2(−1366/+18)-Rev	AAGCTTTTTACCCGGAGCATTCCCG	
Luc3(−890/+18)-For	GGTACCGTTAACTTTGCATAGAGCAGCTTCG	
Luc3(−890/+18)-Rev	AAGCTTTTTACCCGGAGCATTCCCG	
Luc4(−444/+18)-For	GGTACCAGAAAACATGTAAACGCGCCT	
Luc4(−444/+18)-Rev	AAGCTTTTTACCCGGAGCATTCCCG	
Luc5(−343/+18)-For	GGTACCCCGGGCCCTTCCCAGCGCT	
Luc5(−343/+18)-Rev	AAGCTTTTTACCCGGAGCATTCCCG	
Luc6(−235/+18)-For	GGTACCGCGGAAATGACAGTGTGGTG	
Luc6(−235/+18)-Rev	AAGCTTTTTACCCGGAGCATTCCCG	
Luc7(−152/+18)-For	GGTACCAGATCCAGCCGCGGTAGC	
Luc7(−152/+18)-Rev	AAGCTTTTTACCCGGAGCATTCCCG	
Luc5-Mut-NF-κB-For	CTCCCGCTCTGCGGCGGGGGAAATGACAGTGTGGT	site-directed
Luc5-Mut-NF-κB-Rev	ACCACACTGTCATTTCCCCCGCCGCAGAGCGGGAG	
pMID2-For	GGGAATGCTCCGGGTGAA	RT-qPCR
pMID2-Rev	CAGGAGGGGGTCTTCAAACA	
pGAPDH-For	ACATGGCCTCCAAGGAGTAAGA	
pGAPDH-Rev	GATCGAGTTGGGGCTGTGACT	

**Table 2 animals-13-02853-t002:** Sequences of siRNAs.

siRNA	Sequence (5′-3′)
NC-For	UUCUCCGAACGUGUCACGUTT
NC-Rev	ACGUGACACGUUCGGAGAATT
siMID2-For	GGCAACUGCAUCUUCUCAATT
siMID2-Rev	UUGAGAAGAUGCAGUUGCCTT

## Data Availability

The data that support the findings of this study are available from the corresponding author upon reasonable request.

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
