# Peer review of "The Molecular and Function Characterization of Porcine MID2"

_animals, 2023, doi:10.3390/ani13182853_

Round 1
Reviewer 1 Report
This manuscript is written clearly. The results were described in detail. I feel that it can be published with minor changes, mostly in editing of English language. I would like to send the PDF file, where I marked the words and sentences. So that the authors can be read them and change them.

It is included in the comments above.
Author Response
Dear Reviewer:
Thank you very much for your comments and professional advice for our manuscript entitled “The Molecular and Function Characterization of the Porcine MID2 ” (Manuscript ID: animals-2561879). They are very helpful for revising and improving our paper. Based on your suggestions and requests, we have made modifications and highlighted them in red in the revised manuscript.

Reviewer 2 Report
In this manuscript authored by Chen et al., they addressed the pMID2 from pig was cloned and identified for the first time, and the molecular characterization was analyzed by bioinformatics. Then the author firstly evidenced that the NF-κB may be a critical transcription factor for regulating MID2 expression. The data furtherly demonstrated that pMID2 primarily expressed in the porcine lung and spleen, and pMID2 enhanced poly (I: C) and IFN-ɑ-induced JAK-STAT signaling pathway. Their findings implicate pMID2 might act as a positive regulator in porcine immune system. A number of issues are outlined below that if addressed would improve the manuscript overall.
Point 1: Section 2.3. RNA isolation and RT-qPCRs analysis– Mention amount of RNA used.
Point 2: Section 2.4. Plasmids construction – How were the PAMs isolated? The sentence in Line136 and 140 is duplication.
Point 3: Section 2.8. Statistical analysis – Name of the statistical platform is missing; “P” should be italic.
Point 4: Section 2.7. Dual-luciferase reporter assay – Detection of the activation of MID2 promoter and ISRE promoter should be separately written. Line 165, “IFN-β-Luc” should be replaced “ISRE-Luc”. Check again.
Point 5: Section 3.4. Functional promoter analysis of pMID2 – Line 377-378 and line 420-422, the treatment of NF-kB inhibitors is inconsistent, please confirm carefully.
Point 6: Section 3.6. Positive regulatory of JAK-STAT signaling pathway by pMID2 –Line 514,517, the “C” should be replaced “E”. Check again.
Point 7: The manuscript should be checked for grammar: please review and edit.
Author Response
Point 1: Section 2.3. RNA isolation and RT-qPCRs analysis– Mention amount of RNA used.
Response 1: We have added the corresponding contents in Line 120-121 of the revised manuscript.
Point 2: Section 2.4. Plasmids construction – How were the PAMs isolated? The sentence in Line 136 and 140 is duplication.
Response 2: Thanks for your question. We have added the method for the isolation of PAMs in Line 97-98 of the revised manuscript and deleted the duplicate part in Line 136.
Point 3: Section 2.8. Statistical analysis – Name of the statistical platform is missing; “P” should be italic.
Response 3: We have supplemented the relevant content in Line 178-179 of the revised manuscript.
Point 4: Section 2.7. Dual-luciferase reporter assay – Detection of the activation of MID2 promoter and ISRE promoter should be separately written. Line 165, “IFN-β-Luc” should be replaced “ISRE-Luc”. Check again.
Response 4: Sorry for our unclear description. We have described the detection of MID2 promoter and ISRE promoter activation separately (Line 167-175) and replaced “IFN-β-Luc” to “ISRE-Luc” in the revised manuscript.
Point 5: Section 3.4. Functional promoter analysis of pMID2 – Line 377-378 and line 420-422, the treatment of NF-kB inhibitors is inconsistent, please confirm carefully.
Response 5: Thank you for your reminder, we have rewritten this section in Line 354-357 of the revised manuscript.
Point 6: Section 3.6. Positive regulatory of JAK-STAT signaling pathway by pMID2 –Line 514,517, the “C” should be replaced “E”. Check again.
Response 6: Sorry for our negligence.We have corrected it in the revised manuscript.
Point 7: The manuscript should be checked for grammar: please review and edit.
Response 7: Thank you for your suggestions. We apologize for the grammar and spelling issue in the original manuscript, and have made modifications and highlighted them in red in the revised manuscript.

Reviewer 3 Report
The manuscript is providing interesting results that can help elucidate diseases and infections of pigs.
The findings indicate significant benefits for the treatment of pigs and should be publicized.
I support publication, but careful revision is necessary before final acceptance.
Major issues.
-Simple summary is not a simple summary, as it contains many terms that are difficult for lay people to grasp. The section requires complete rewriting from the start.
-Three animals is a really small number and, although I understand the limitations, the authors should justify this small number in the discussion.
Minor issues
-The objectives of the study must be described clearly.
-You need to include a paragraph about infections of pigs, in order to make readers familiar with the clinical topic of the study.
-Table 1. You need to include all the details of the PCR, not just the primers.
-Figure 1 B. You can move this to appendix.
-Discussion (1): please include a new section to discuss the clinical applications of these findings and also how do you expect to commercialize the findings.
-Discussion (2): please add a brief passage to discuss also the benefits of the findings in neoplastic diseases of pigs (although this is not the primary scope of the work).
General. The study is a worthy manuscript and merits publication after revision as indicated.
Author Response
Point 1:Simple summary is not a simple summary, as it contains many terms that are difficult for lay people to grasp. The section requires complete rewriting from the start.
Response 1: Thank you for your suggestion. We have rewritten the content of this section according to the reviewer’s comment, line 13-21.
Point 2:Three animals is a really small number and, although I understand the limitations, the authors should justify this small number in the discussion.
Response 2: We have added the corresponding contents to the discussion of the revised manuscript (line 484-485).
Point 3:The objectives of the study must be described clearly.
Response 3: According to reviewer’s comment, we have modified it in revised manuscript (line 35-38).
Point 4:You need to include a paragraph about infections of pigs, in order to make readers familiar with the clinical topic of the study.
Response 4: Thanks for your suggestion. We are exploring relevant research and deeply investigating the correlation of infected pigs and MID2.
Point 5:Table 1. You need to include all the details of the PCR, not just the primers.
Response 5: Thanks for your comment. In the revised manuscript, we have added the details of the PCR reaction (Line130-132).
Point 6:Figure 1 B. You can move this to appendix.
Response 6: According to reviewer’s comment, we have changed “Figure 1B” to “supplement Figure 1( Figure S1)” in the revised manuscript.
Point 7:Discussion (1): please include a new section to discuss the clinical applications of these findings and also how do you expect to commercialize the findings.
-Discussion (2): please add a brief passage to discuss also the benefits of the findings in neoplastic diseases of pigs (although this is not the primary scope of the work).
Response 7: We are grateful for the suggestion. On the recommendation of the reviewers, we have added a discussion of the potential impact of this study on clinical applications, commercial value (Line 553-557), and porcine neoplastic diseases (Line 536-540) to the revised manuscript.

Round 2
Reviewer 3 Report
The manuscript has been improved.
The only comment that I have is that the simple summary still includes many technical terms, which cannot be grasped by lay persons. Please modify the simple summary to make it understood by people who are not scientists, this is the real meaning of the passage, so this should not be a more simple abstract.
After making this change, the manuscript can be accepted.
Author Response
Thank you for your comments and professional advice, we have made modifications to the content of this section in the revised manuscript.
